# Molecular Mechanisms of the Stripe Rust Interaction with Resistant and Susceptible Wheat Genotypes

**DOI:** 10.3390/ijms25052930

**Published:** 2024-03-02

**Authors:** Taras Nazarov, Yan Liu, Xianming Chen, Deven R. See

**Affiliations:** 1Department of Plant Pathology, Washington State University, Pullman, WA 99164-6430, USA; taras.nazarov@wsu.edu (T.N.); y.liu@wsu.edu (Y.L.); xianming.chen@usda.gov (X.C.); 2US Department of Agriculture, Agricultural Research Service, Wheat Health, Genetics, and Quality Research Unit, Pullman, WA 99164-6430, USA

**Keywords:** wheat, stripe rust, host–pathogen interaction, RNA-seq, transcriptome

## Abstract

Rust fungi cause significant damage to wheat production worldwide. In order to mitigate disease impact and improve food security via durable resistance, it is important to understand the molecular basis of host–pathogen interactions. Despite a long history of research and high agricultural importance, still little is known about the interactions between the stripe rust fungus and wheat host on the gene expression level. Here, we present analysis of the molecular interactions between a major wheat pathogen—*Puccinia striiformis* f. sp. *tritici* (*Pst*)—in resistant and susceptible host backgrounds. Using plants with durable nonrace-specific resistance along with fully susceptible ones allowed us to show how gene expression patterns shift in compatible versus incompatible interactions. The pathogen showed significantly greater number and fold changes of overexpressed genes on the resistant host than the susceptible host. Stress-related pathways including MAPK, oxidation–reduction, osmotic stress, and stress granule formation were, almost exclusively, upregulated in the resistant host background, suggesting the requirement of the resistance-countermeasure mechanism facilitated by *Pst*. In contrast, the susceptible host background allowed for broad overrepresentation of the nutrient uptake pathways. This is the first study focused on the stripe rust pathogen–wheat interactions, on the whole transcriptome level, from the pathogen side. It lays a foundation for the better understanding of the resistant/susceptible hosts versus pathogenic fungus interaction in a broader sense.

## 1. Introduction

Wheat is a major food source for the human population and the second most produced cereal crop in the world (https://www.statista.com/statistics/263977/world-grain-production-by-type/, accessed on 8 January 2024). It provides more than 20% of the protein and calories for the world’s population and holds the top position for global crop trade and amount of land used for its production [1]. Being one of the first domesticated crops, wheat has approximately a 10,000-year history of selection and breeding improvements [2]. Despite such a long history of agronomic and breeding practice, the demand for effective wheat production still poses significant scientific and technological challenges. Along with climatic challenges, biotic stress, including bacterial, viral, and fungal pathogens, is a major constraint to wheat production.

Leaf, stem, and stripe rusts of wheat, caused by *Puccinia triticina* (*Pt*), *P. graminis* f. sp. *tritici* (*Pgt*), and *P. striiformis* f. sp. *tritici* (*Pst*), respectively, are major wheat pathogens associated with regular and significant yield losses [3]. Despite decades of systematic work on resistance genetics and breeding, rust remains a major biotic threat to global wheat production [4]. Of all wheat rust diseases, stripe rust occurs most frequently in the United States, mainly due to favorable conditions for early infection, long-distance spore dispersal, and possible recent adaptation to warmer temperatures. It is especially destructive in temperate and humid wheat growing regions and characterized by yield losses up to 100% in susceptible cultivars [5]. Although fungicide applications to prevent yield losses are widely used and effective, they are expensive and have negative environmental impact. Genetic resistance is a more economical and environmentally friendly way to control stripe rust. 

There are two major types of genetic resistance to stripe rust in wheat: race-specific all-stage resistance (ASR) and adult plant resistance (APR). ASR is easy to detect in seedling tests and it remains effective through all stages of the plant lifecycle. It is usually conferred by single genes with strong effects. Widely adopted by breeders during the 20th century, these genes were the fastest way to introduce resistance into wheat cultivars. However, introduction of such strong resistance genes increased selective pressure on the pathogen and led to multiple cases of rapid emergence of the highly virulent *Pst* races. APR, on the other hand, provides broad-spectrum defense, while compromising on the degree of response. Plants carrying APR genes or QTL are susceptible in seedling tests but express varying levels of resistance in post-seedling stages in both field and greenhouse environments [6]. One of the most efficient types of APR is high-temperature adult-plant resistance (HTAP), which is durable, nonrace-specific, and triggered in the late developmental stages [7].

Studying the mechanisms of interaction between highly virulent *Pst* races and durable resistant germplasms could lead to a better understanding of factors influencing resistance and aid in elucidation of biotrophic pathogenic processes. Simultaneous transcriptome sequencing of plant and pathogen can provide insights into genome-wide *Pst*–wheat interactions [8,9,10,11,12]. Several attempts have been made, up to date, to understand stripe rust gene expression: urediniospores cDNA library characterization [13], isolated haustoria and infected tissue analysis [14], germinated urediniospores versus haustorial transcriptome [15], and microarray transcript analysis of compatible versus incompatible *Pst*–wheat interactions [16], as well as the analysis of whole-genome transcriptomes of both wheat and stripe rust pathogen during infection using a dual RNA sequencing (RNA-seq) method [17,18]. 

Few findings from studying rust expression patterns suggest that fungus depends on host metabolism via an expanded repertoire of amino acid and peptide transporters along with loss of nitrate and sulfate assimilation pathways [19]. The most commonly predicted upregulated genes are associated with rust colonization code for energy production and hydrolytic enzymes, effector-like secreted proteins, and other proteins of unknown functions. Many of the latter proteins are specific to different rust fungus species. Furthermore, the majority of rust genes do not have homologs with known functions in the GO database. Among known homologues, there are few functional groups, including transporters, kinases, carbohydrate-active enzymes (CAZY), and secreted proteins (SPs), which are represented by higher proportion compared to other genes. This overrepresentation of SPs is consistent with observations in other obligate biotrophic plant pathogenic fungi [17,20].

Despite discoveries of expression levels and gene families for specific stages of the pathogen lifecycle, there are no reports on the whole *Pst* transcriptome, affected by durable broad-range-resistant versus susceptible host germplasms. Here, we present gene expression analysis of the highly virulent *Pst* race PST-100 using the next-generation sequencing method. To understand the influence of effective plant resistance on rust pathogen development, we used combined transcriptome from interactions between *Pst* and resistant versus susceptible wheat cultivars. Importantly, this study was aimed at better understanding of the mechanisms of wheat–*Pst* interactions by determining gene expression patterns and identifying upregulated/downregulated targets in an economically important HTAP resistance host background.

## 2. Results and Discussion

### 2.1. Sequencing, Mapping, and Expression Profile

A total of 6.6 million reads were obtained from both treatments, including the resistant and susceptible bulks, with 3.6 million reads for resistant bulk and 3.0 million reads for susceptible bulk (Appendix A). Out of both transcriptome sets, 10% of resistant and 11% of susceptible reads were mapped to the PST-78 reference genome. In total, 6791 genes showed significant expression with more than 10 TPM (transcripts per million), and 3425 of those genes were significantly expressed in both treatments. Out of total genes, 3366 showed differential expression with more than a 2-fold change between treatments; 2808 overexpressed in the resistant background and 558 overexpressed in the susceptible background; 1039 genes were unique (not expressed in another dataset) in the resistant background and 269 genes in susceptible background (Figure 1). The average fold change for the unique genes, expressed in the resistant background, was 3.8, ranging from 2.5 to 28. For the unique genes, expressed in the susceptible background, the average fold change was 3.2, ranging from 2.4 to 42. There were 69 genes with >10-fold upregulation (maximum fold change is 45) from resistant background and 4 genes with >10-fold upregulation (maximum fold change is 81) from susceptible background. The top 20 upregulated genes for each background are presented in Table 1 and Table 2, respectively.

In this study, five times the number of overexpressed genes and almost four times the number of uniquely expressed genes were induced in rust pathogen by resistant wheat host compared to the susceptible wheat host. Previous studies also indicated that greater numbers of genes were induced in resistant wheat plants in response to wheat rust infection [21,22,23]. Our findings indicate that an even broader repertoire of genes was induced in the resistant bulk wheat lines carrying nonspecific HTAP resistance, compared to the race-specific, all-stage resistant. These results may suggest that the observed upregulation pattern could be the result of the efficient plant resistance response facilitated via multiple resistance genes activation, which, in turn, induce coping mechanisms from the fungus side in a form of the broader expression of the defense- and pathogenicity-related genes. 

### 2.2. Annotation Summary

Among 3366 differentially expressed genes, 3364 had significant BLAST scores (e-value cutoff = 1.0 × 10^−5^) and 1961 mapped to GO terms, out of which 1833 were functionally annotated. A total of 99.5% transcripts of the top BLAST hits were from the *Pst* genome. Overall, the 4 species with the most hits were *Pst* (48%), *Puccinia graminis* f. sp. *Tritici* (21%), *Puccinia sorghi* (13%), and *Melampsora larici-populina* (9%) (Figure 2). Notably, along with 99.5% of the top hits belonging to the same species, 98% of total hits belonged to the order of Pucciniales, suggesting a highly specialized repertoire of gene expression specific to members of this order. Indeed, previous observations suggested large proportions of species-, family-, and order-specific candidate secreted effector proteins in rust fungi [24,25].

### 2.3. Commonly Expressed Genes

Out of 3425 *Pst* genes significantly expressed in both host conditions, 2332 did not show significantly differential expression (<2-fold change). Common for both backgrounds, the *Pst* genes with the highest level of expression (>100 TPM) belonged to cellular and metabolic processes, regulation, localization, and response to stimulus in GO terms for biological process (Figure 3). Genes with the most specific annotations were related to ribosome biogenesis, translation, protein folding, ubiquitin-related catabolism, and transcriptional regulation. Most general annotations for common molecular functions were represented by binding (ATP, nucleic acid, metal, GTP), catalytic activity, and structural molecule activity (Figure 3).

### 2.4. Differentially Expressed Genes

The most highly upregulated *Pst* genes in the susceptible background, compared to the resistant background, were PSTG_02354 and PSTG_03920, with 81- and 42-fold upregulation, respectively. Both genes code for putative proteins with unknown functions without similarity in InterPro databases (Table 2). In the resistant background, the most upregulated *Pst* genes were PSTG_08955, PSTG_06581, PSTG_00606, PSTG_12286, PSTG_03460, and PSTG_02011, ranging from 45- to 19-fold changes. All of them code for hypothetical proteins, with PSTG_08955 (the most upregulated) and PSTG_12286 having no further annotation. PSTG_06581, PSTG_00606, PSTG_03460, and PSTG_02011 were annotated as integral components of membranes. PSTG_03460 is associated with protein transport and PSTG_02011 with the oxidation–reduction process (Table 1).

The putative *Pst* gene PSTG_02011, which was highly (19-fold) upregulated and involved in oxidative stress response (OSR) in the resistant host background, indicates expectedly higher pressure on the pathogen. Along with the well-established role of reactive oxygen species (ROS) in plant defense response [26], there is a growing body of evidence for ROS-associated OSR importance for the pathogenic fungi, especially in the initial stage of the infection [27,28]. 

Another putative *Pst* gene, PSTG_02239, with 15-fold upregulation in the resistant host background, was annotated as thyroid receptor, which interacts and was involved in palmitoyltransferase activity. Although its function is unknown, previous findings suggest that at least one fungus (*Glomus intraradices*) codes for mammal-like thyroid-interacting protein, which belongs to archetypal regulatory proteins involved in intracellular hormonal signaling [29]. Additionally, palmitoyltransferases have been reported to play an important role for hyphal morphogenesis, cell wall integrity, and virulence of *Aspergillus fumigatus* [30].

The only annotated and highly upregulated putative gene in susceptible background compared to resistant background was PSTG_18287, with a 12-fold increase, which codes for ATP synthase.

Several putative genes, which were among the most upregulated in both resistant and susceptible backgrounds, were annotated as integral membrane components. In order to understand their functions, further investigation is needed, since they can play diverse roles from the involvement in signaling and effector secretion in the initial infection phase to hexose transport metabolism after successful colonization [31].

### 2.5. Overrepresentation Analysis

To estimate the difference between two host backgrounds on the systemic level, we performed two-sided Fisher’s exact test on predicted GO terms. Predicted molecular functions of the genes overrepresented in the susceptible background include structural constituent of ribosome, structural molecule activity, ubiquitin–protein transferase activity, and transferase activity (transferring acyl groups). In the resistant background, the most overrepresented molecular function of genes was binding, which includes organic cyclic compound binding, heterocyclic compound binding, macromolecular complex binding, and identical proteins binding (Table 3). 

The most abundantly overrepresented biological processes in the susceptible background include sulfur compound metabolic process, cytoplasmic translation, cofactor metabolic process, sulfur compound biosynthetic process, coenzyme metabolic process, monocarboxylic acid metabolic process, monosaccharide biosynthetic process, hexose biosynthetic process, glucose metabolic process, gluconeogenesis, peptide metabolic process, ribosome biogenesis, translation, cofactor biosynthetic process, and nucleus organization. In the resistant background, the most overrepresented biological processes include regulation of molecular functions, regulation of catalytic activity, cellular response to stimulus, nucleic acid metabolic process, regulation of hydrolase activity, positive regulation of catalytic activity, molecular function, and hydrolase activity (Table 4).

We hypothesize that overrepresentation of monosaccharide biosynthetic process, hexose biosynthetic process, glucose metabolic process, and gluconeogenesis in susceptible plants is a result of successfully established pathogenicity and hyphal proliferation [17,32], which allows fungus to induce a broader repertoire of feeding-related pathways. On the other hand, overrepresentation of hydrolase-related pathways in the resistant wheat background could be an indication of an additional need in a cell wall degrading machinery since resistant wheat activates penetration-protective mechanisms via phenylalanine ammonia-lyase-induced lignin production [16].

### 2.6. Enzyme Profile

Enzyme coding genes distribution did not show significant differences between two host backgrounds except for relative higher abundance of isomerases and ligases in the susceptible dataset and transferases in the resistant dataset (Figure 4). The total percentages of predicted enzymes were 17.8 and 13.6 for the resistant and susceptible background, respectively. The three largest classes of the enzymes for both conditions were hydrolases, with 44.5% and 42.1% for the resistant- and susceptible-host-associated genes, followed by transferases 30.9% and 26.3%, and oxidoreductases 13.4% and 11.8%, respectively. Lyases, isomerases, and ligases comprised 4.2%, 3.4%, and 3.6% for the resistance-associated genes and 5.3%, 7.9%, and 6.6% for the susceptible set. 

### 2.7. Stress Response

To compare stress effects of potentially more unfavorable resistant versus susceptible host background, we ran a multilevel annotation search on stress-related GO terms (Table 5). Out of 39 stress-related genes, 34 were upregulated in the resistant background. Fold change was also greater in the resistant background, with 11 genes showing >5-fold upregulation (9.5 max), while in the susceptible background, upregulation fold ranged from 2.4 to 4.8. PSTG_14185 was the most upregulated putative *Pst* gene in the resistant background, coding for PAKA kinase, associated with an MAPK stress response cascade. PAKAs, or p21-activated kinases, have a wide range of cellular functions, including a control of cytoskeletal organization, cell growth, and cell survival [33]. Specifically, PAKA kinases are reported as a major component in ROS scavenging in the grass pathogen *Claviceps purpurea* [34]. Along with three other putative *Pst* genes (PSTG_16900, PSTG_09637, and PSTG_00069) that were upregulated in the resistant background, PSTG_14185 is a part of an MAPK cascade which is reported to play an important role in the establishment of various infection strategies for plant pathogenic fungi [35,36]. We hypothesize that upregulation of such genes could be, partially, due to the initial response from the resistant host, which prevents fungal penetration and rapid establishment of feeding structures. Indeed, MAPK pathways have been reported to play an important role for appressorial formation in *Cochliobolus heterostrophus* [37], *Colletotrichum orbiculare* [38,39,40], and *Pyrenophora teres* [41]. An MAPK expression is also required for the induction of cellulase-encoding genes and controlling host tissue penetration [42]. Interestingly, fungicide treatment, in addition to osmotic and oxidative stress, have been reported to activate MAPK pathways in some plant pathogenic fungi: *Cochliobolus heterostrophus*, *Neurospora crassa* [43], and *Botrytis cinerea* [44]. High upregulation levels of *Pst* MAPK-associated genes in the resistant plants from our experiments could be an indication of the compensatory reaction to the efficient plant pattern triggered immunity (PTI). It might work in, at least, two directions: to form more penetration structures and to activate ROS scavenging mechanisms for the prevention of further damage to the pathogen. 

PSTG_07441 is the other putative *Pst* gene upregulated in the resistant host that could play a counterdefense role against plant initial PTI. It showed a 6.3-fold increase and belongs to the ABC transporter family. Along with a predicted roles in active transmembrane transport, ion channels, and receptor functions; [45,46,47], ABC transporters play roles in coping with host-plant-induced cytotoxicity and oxidative stress within appressoria during early stages of infection in *Magnaporthe grisea* [48]. 

A set of putative genes including PSTG_01011, PSTG_11190, PSTG_02449, and PSTG_07071 related to stress granule formation were upregulated exclusively in the resistant host. Although functionality of stress granules is poorly understood, they are reported to be formed in response to stress and generally are not observable under normal growth [49]. Furthermore, stress granule formation is related to endoplasmic reticulum (ER), oxidative, and osmotic stresses, and plays an important role in the survival of *Aspergillus oryzae* cells exposed to stress [50]. Overexpression of the genes related to stress granule formation only in the resistant host might be an additional indicator of PTI-induced stress coping reaction that is not required in the case of successful colonization of the susceptible plant. 

The majority of putative stress-related genes belonged to oxidative stress response pathways. Out of 16 differentially expressed OSR related genes, 13 were upregulated in the resistant host, which included PSTG_03524, PSTG_07441, PSTG_15614, PSTG_06116, PSTG_06546, PSTG_11344, PSTG_12053, PSTG_00788, PSTG_16670, PSTG_07945, PSTG_02845, PSTG_07189, and PSTG_09261. Although the number of such genes was greater in the resistant plant background, three out of five stress-related genes upregulated in the susceptible host, PSTG_20181, PSTG_12250, and PSTG_10795, were also related to oxidative stress response. The oxidative burst is widely reported as a basal plant defense against pathogens [51,52]. It is one of the fastest and the most ubiquitous PAMP-recognition-triggered responses. A major share of the ROS-related pathway upregulation in our experiments aligns with the previous findings that it is directly related to fungal pathogenicity metabolic processes [27]. It serves as a source of pathogen-produced oxidative stress, defense reaction, and signaling to induce cell differentiation as a part of a colonization strategy [53]. Due to the ubiquitous nature of the oxidative burst as a basal plant defense, genes related to ROS in our experiments were upregulated in both resistant and susceptible host backgrounds, suggesting a quantitative nature of the initial resistance reaction from the plant side. Despite greater proportion of ROS-related genes upregulated in the susceptible host background, the resistant plants triggered higher fold changes and total number of such genes.

Several putative genes related to salt and osmotic stress response were upregulated in the resistant plant background, which were PSTG_06288, PSTG_07422, PSTG_11998, PSTG_13165, and PSTG_09157, and one gene, PSTG_05154, in susceptible plants. Osmotic pressure stress affects fungi upon cell wall lysis and plant cell penetration; additionally, osmotic stress response is intertwined with the MAP kinase signaling pathway and the high-osmolarity glycerol (HOG) pathway [54]. Combined with OSR, they comprise a broad network of stress responses [55]. Expectedly putative *Pst* genes related to osmotic stress response were expressed in both experimental conditions, although a greater number was observed in the resistant host. A possible combination of a stronger oxidative burst response from the resistant plants and OSR/osmotic stress response pathway contributed to the observed upregulation.

## 3. Materials and Methods

### 3.1. Host and Pathogen Materials

A population of F_5:6_ spring wheat (*Triticum aestivum* L.) recombinant inbred lines (RILs) was used as the host material resources [56]. The seeds of the RILs were provided by the winter wheat breeding program, department of crop and soil science, Washington State University, Pullman, WA USA. The RILs, including 188 individuals, were developed from a single F_1_ plant derived from the cross of Louise (PI 634865) and Penawawa using the single-seed descent method. Penawawa, a soft white spring cultivar, shows susceptibility to most current races of *Pst*, while a soft spring wheat cultivar, Louise, carries a potentially novel HTAP gene for stripe rust resistance [56]. Such selection of host materials allowed us to design experiments with both compatible and incompatible interactions between wheat and *Pst*. Stripe rust race, PST-100, was used as a fungal component which was preserved and reproduced following the standard procedure in the wheat stripe rust research lab at USDA ARS, Pullman WA [57]. It is highly virulent and the most distributed *Pst* race in the US in recent years [58]. 

### 3.2. Greenhouse Experiments

Two bulk experimental sets comprised 11 resistant RILs and 10 susceptible RILs which were selected from 188 F_5:6_ RILs based on their HTAP reactions in the field, inoculated with PST-100 for compatible and incompatible interactions, respectively. Each experimental treatment had 3 replicates to normalize for gene expression analysis. Eight seeds of each RIL were planted in a round gallon pot of 15 cm in diameter and grown in a greenhouse with a diurnal cycle: 16 h light at 25 °C; 8 h dark at 15 °C. After 42 days, plants with fully emerged flag leaf (Feekes stage 9) were inoculated with a urediniospore/talc mixture (1:10 ratio) following the standard procedure of inoculating [57]. Plants were sprayed with sterile water, and the urediniospore/talc mixture was evenly applied to both sides of the flag leaf using a cotton swap. Control plants underwent the same process except for the absence of the urediniospores in talc application. To promote effective spore germination and penetration, plants were placed for 24 h in a dew chamber set to 10 °C and 100% relative humidity in dark. Plants were subsequently incubated in a growth chamber, set for diurnal cycles of 16 h light at 25 ± 1 °C, and 8 h dark at 15 °C. 

### 3.3. Library Preparation and Sequencing

Flag leaves of each RIL were collected at 48 h post inoculation and instantly frozen using liquid nitrogen. Such exposure time showed a peak in transcript accumulation, associated with HTAP resistance. A total of 63 samples (21 lines × 3 replicates, 8 leaves per replicate) were collected for analysis. Total RNA was extracted with TRIzol^®^ Reagent (Thermo Fisher Scientific, Carlsbad, CA, USA) using triplicated combined tissue from each inoculated RIL. MicroPoly(A)Purist™ mRNA purification kit and Dynabeads^®^ mRNA DIRECT™ Purification Kit (Thermo Fisher Scientific, Carlsbad, CA, USA) were used to isolate mRNA from total RNA. Equal quantities of purified RNA samples from 11 resistant and 10 susceptible lines were pooled to create two bulk sets, respectively. RNA-seq libraries were constructed using Ion Total RNA-Seq Kit and Ion Total RNA-Seq Kit v2 (Thermo Fisher Scientific, Carlsbad, CA, USA). Libraries were barcoded using Ion Xpress™ RNA-Seq Barcode 1-16 Kit and sequenced on the Ion Torrent PGM™ semiconductor sequencer (Thermo Fisher Scientific, Carlsbad, CA, USA) with Ion 318™ chips at USDA ARS Western Regional Small Grains Genotyping laboratory, Pullman, WA, USA.

### 3.4. Bioinformatics Pipeline

Initial raw read processing, including trimming of adaptors, barcodes, and library-specific separation of barcoded reads, was performed using Ion Torrent Browser software 5.2 (Thermo Fisher Scientific, Carlsbad, CA, USA). Trimmed, library-specific reads were exported to CLC Genomics Workbench (https://digitalinsights.qiagen.com, accessed on May 2016) with further quality trimming (PHRED score to error probability = 0.05) and mapped to the PST-78 genome (*Puccinia* Group Sequencing Project, Broad Institute of Harvard and MIT). PST-78 supercontigs, genes, and mRNA tracks were used for the mapping, with the following settings: mismatch cost = 2; insertion cost = 3; deletion cost = 3; minimum alignment length fraction = 0.8; similarity fraction = 0.8; and 5 maximum allowed hits for a read. Expression values were calculated in TPM—transcripts per million mapped reads [59] and normalized using Baggerly’s test [60,61]. Differential expression analysis performed on resistant versus susceptible reaction sets with minimum fold change ≥ 2 and FDR corrected *p*-value < 0.05.

Annotation was performed using NCBI blastx (https://www.ncbi.nlm.nih.gov/, accessed on May 2016 ) of the total nonredundant protein sequence database with e-value 1.0 × 10^−5^ and 5 best hits parameters. EBI InterProScan was run on blast results using BlastProDom, FPrintScan, HMMPIR, HMMPfam, HMMSmart, HMMTigr, ProfileScan, PatternScan, SuperFamily, HMMPanther, and Gene3dD components in Blast2GO PRO (https://www.blast2go.com/, accessed on May 2016). Gene Ontology terms were assigned using NCBI, PIR, and GO databases. GO terms were annotated using the following parameters: annotation cutoff = 55; GO weight = 5; computation analysis evidence codes = 0.8 (ISS, ISO, ISA, ISM, IBA, IBD, IKR, RCA) and 0.7 (IGC, IRD); and experimental evidence codes = 1. 

Overrepresentation analysis for the annotated transcripts was performed with two-sided Fisher’s exact test on predicted GO terms with *p*-value cutoff = 0.05. 

## 4. Conclusions

This study presents a comparative analysis of *Pst* gene expression in partially resistant and susceptible wheat cultivars. Applying next-generation transcriptomics allowed us to show genome-wide *Pst* differential expression patterns, while overcoming restrictions of the previous microarray and cDNA-AFLP studies [62,63], which were limited to the already-known probes or polymorphism. The analysis of both gene count and upregulation level shows higher levels of *Pst* gene expression in the resistant host background. An observed 5:1 ratio of the significantly expressed putative *Pst* genes in the resistant versus susceptible host indicates a possible need of the plant resistance-countermeasure mechanisms for the fungus functionality. This is supported by an even greater (7:1) ratio of the stress-related genes induced by the resistant plants. A broad repertoire of stress-related coping responses included MAPKs, oxidation stress reduction, osmotic stress, and stress granule formation pathways. In addition, hydrolase production pathways were also overrepresented in the resistant background, suggesting that auxiliary requirements to mitigate cell wall reinforcement machinery were upregulated in resistant plants. The susceptible reaction, on the other hand, induced overrepresentation of the several nutrient-uptake-related pathways, indicating effective establishment of the pathogenicity. The most upregulated genes from both conditions did not provide any insight about their functions, suggesting the need for further investigation. Overall, the results of this study lay a foundation for a better understanding of the wheat–*Pst* interactions, from the pathogen side, especially mediated by durable plant resistance.

## Figures and Tables

**Figure 1 ijms-25-02930-f001:**
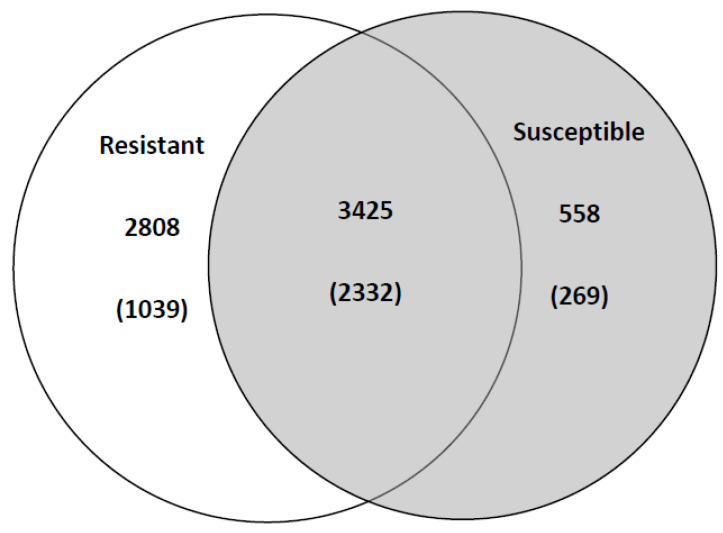
Differentially expressed PST genes. The white circle represents *Puccinia striiformis* f. sp. *tritici* (PST) genes significantly (>10 TPM) and differentially (>2 fold) expressed in resistant host background, while the grey circle represents the differentially expressed PST genes in susceptible background. Numbers in parentheses represent uniquely expressed PST genes (expressed only in one condition). The intersection represents significantly expressed PST genes (>10 TPM) in both host backgrounds and the number in parentheses represents PST genes without significantly differential expression in both backgrounds (<2 fold).

**Figure 2 ijms-25-02930-f002:**
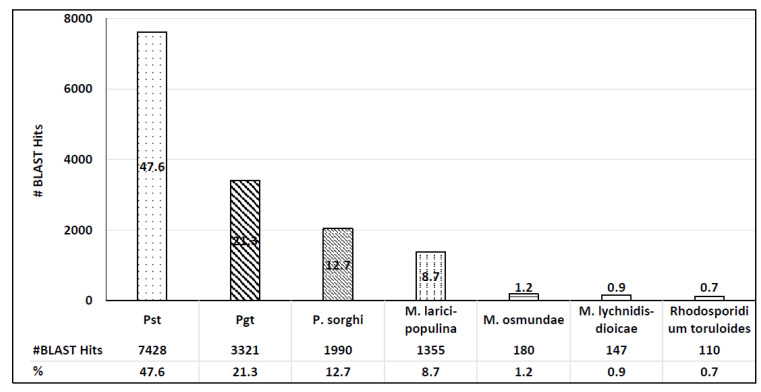
The distribution of blast hits per species. The Y axis represents total number of blast hits, and the X axis represents species in descending order by number of hits.

**Figure 3 ijms-25-02930-f003:**
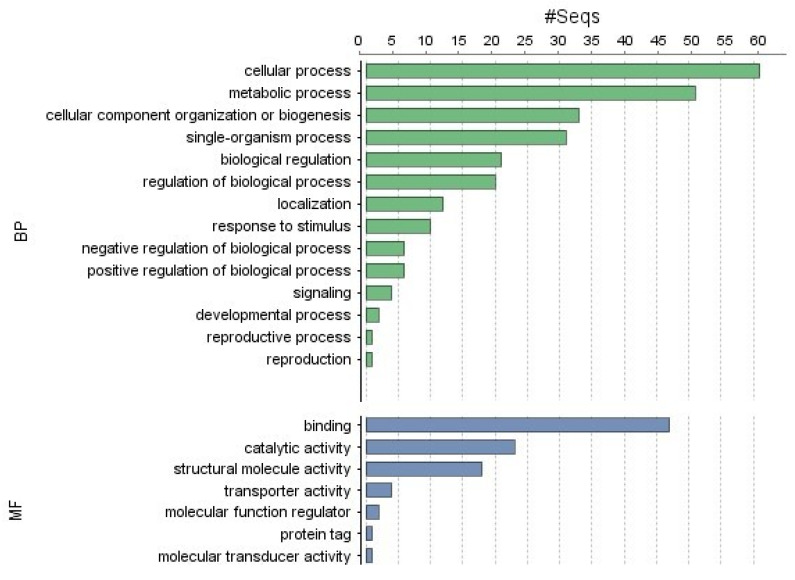
The annotation of commonly differential expressed PST genes. The top section represents the annotation for biological process (BP). The bottom section represents the annotation for molecular function (MF). Terms arranged in descending order by number of sequences shown on the X axis. The Y axis describes the annotation terms.

**Figure 4 ijms-25-02930-f004:**
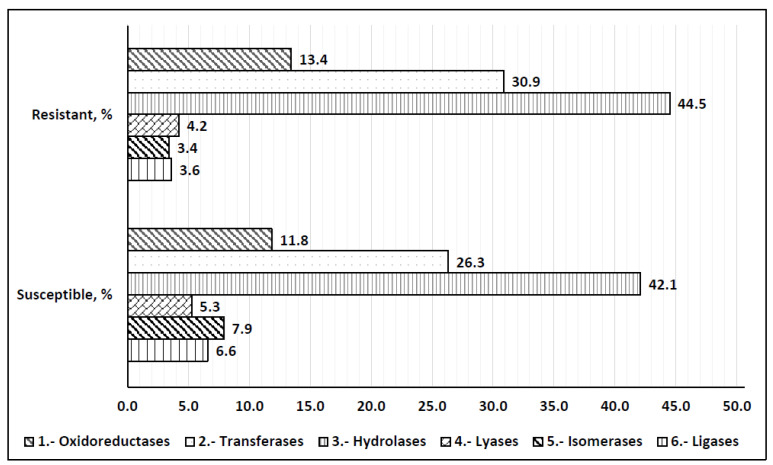
Enzyme coding gene distribution. Upper cluster represents enzyme classes from resistant host associated genes; lower—from susceptible, X axis represents a percentage of a given enzyme.

**Table 1 ijms-25-02930-t001:** The information of the top 20 *Pst* genes that were upregulated in the resistant host background.

Gene ID ^a^	Fold	Description	GO Names List ^b^
PSTG_08955	45	hypothetical protein	
PSTG_06581	28	hypothetical protein	C: integral component of membrane
PSTG_00606	25	hypothetical protein	C: integral component of membrane
PSTG_12286	25	hypothetical protein	
PSTG_03460	20	hypothetical protein	P: protein transport; C: integral component of membrane
PSTG_02011	19	hypothetical protein	C: integral component of membrane; F: oxidoreductase activity, flavin adenine dinucleotide binding; P: oxidation-reduction process
PSTG_15628	16	hypothetical protein	
PSTG_02239	15	thyroid receptor-interacting 13	F: ATPase activity, ATP binding, zinc ion binding, protein-cysteine S-palmitoyltransferase activity; P: acyl-carrier-protein biosynthetic process; C: integral component of membrane
PSTG_19929	15	hypothetical protein	
PSTG_16771	15	hypothetical protein	F: metal ion binding
PSTG_00924	14	[*Laccaria bicolor* S238N-H82]	P: tRNA methylation; F: tRNA (guanine) methyltransferase activity
PSTG_05075	14	hypothetical protein	P: metabolic process
PSTG_08572	14	adenosinetriphosphatase	C: U2-type post-spliceosomal complex; F: ATP binding, poly(A) RNA binding, ATP-dependent RNA helicase activity; P: spliceosomal complex disassembly;
PSTG_12570	14	hypothetical protein	
PSTG_15852	14	hypothetical protein	C: membrane, integral component of membrane
PSTG_16422	14	hypothetical protein	F: GTPase activator activity; P: positive regulation of GTPase activity
PSTG_09704	14	hypothetical protein	
PSTG_06470	13	hypothetical protein	
PSTG_14058	13	hypothetical protein	
PSTG_15653	13	AB-hydrolase associated lipase	C: integral component of membrane; F: hydrolase activity; P: lipid metabolic process;

Note: ^a^: Gene ID, which represents top BLAST result and is arranged via descending fold change order. ^b^: GO names list contains the combined GO terms; C means cellular component, F means molecular function, and P means biological process.

**Table 2 ijms-25-02930-t002:** The information of the top 20 *Pst* genes that were upregulated in the susceptible host background.

Gene ID ^a^	Fold	Description	GO Names List ^b^
PSTG_02354	81	hypothetical protein	
PSTG_03920	42	signal peptide	
PSTG_18287	12	ATP synthase subunit 6	C: integral component of membrane; F: hydrogen ion transmembrane transporter activity; P: ATP synthesis coupled proton transport
PSTG_10842	11	hypothetical protein	
PSTG_19930	10	hypothetical protein	
PSTG_13459	9	hypothetical protein	
PSTG_04402	9	hypothetical protein	C: integral component of membrane; P: transmembrane transport
PSTG_11290	8	hypothetical protein	
PSTG_12513	8	hypothetical protein	
PSTG_15937	8	hypothetical protein	
PSTG_11902	8	hypothetical protein	
PSTG_04973	7	helicase-like 2	C: replication fork; F: ATP binding, ATP-dependent helicase activity; P: box C/D snoRNP assembly, histone exchange, rRNA processing, regulation of transcription
PSTG_08876	7	hypothetical protein	F: phosphatidylinositol-4-phosphate binding, unfolded protein binding, phosphatidic acid binding; P: posttranslational protein targeting to membrane
PSTG_08887	7	hypothetical protein	P: regulation of transcription, DNA-templated
PSTG_11993	7	hypothetical protein	F: hydrolase activity; P: metabolic process
PSTG_01766	6	CSEP-partial	
PSTG_11765	6	hypothetical protein	
PSTG_11806	6	hypothetical protein	
PSTG_12910	6	hypothetical protein	
PSTG_13951	6	hypothetical protein	

Note: ^a^: Gene ID, which represents top BLAST result and arranged via descending fold change order. ^b^: GO names list contains the combined GO terms; C means cellular component, F means molecular function, and P means biological process.

**Table 3 ijms-25-02930-t003:** Molecular function of predicted GO terms of overrepresented genes.

GO-ID	Term	*p*-Value	Background
GO:0003735	structural constituent of ribosome	0.00	Susceptible
GO:0005198	structural molecule activity	0.00	Susceptible
GO:0004842	ubiquitin-protein transferase activity	0.01	Susceptible
GO:0019787	ubiquitin-like protein transferase activity	0.01	Susceptible
GO:0016886	ligase activity, forming phosphoric ester bonds	0.02	Susceptible
GO:0016868	intramolecular transferase activity, phosphotransferases	0.02	Susceptible
GO:1990050	phosphatidic acid transporter activity	0.02	Susceptible
GO:0070273	phosphatidylinositol-4-phosphate binding	0.02	Susceptible
GO:0008242	omega peptidase activity	0.02	Susceptible
GO:0003878	ATP citrate synthase activity	0.02	Susceptible
GO:0046912	transferase activity, transferring acyl groups	0.03	Susceptible
GO:0005488	binding	0.00	Resistant
GO:0097159	organic cyclic compound binding	0.02	Resistant
GO:1901363	heterocyclic compound binding	0.02	Resistant
GO:0044877	macromolecular complex binding	0.04	Resistant
GO:0042802	identical protein binding	0.04	Resistant
GO:0016772	transferase activity, transferring phosphorus-containing groups	0.05	Resistant

Note: Top section represents gene ontology molecular function classes overrepresented in susceptible host background; bottom section represents molecular function in resistant host background. Terms arranged in ascending order by *p*-value.

**Table 4 ijms-25-02930-t004:** Biological process of predicted GO terms of overrepresented genes.

GO ID	Term	*p*-Value	Background
GO:0006790	sulfur compound metabolic process	0.00	Susceptible
GO:0002181	cytoplasmic translation	0.00	Susceptible
GO:0051186	cofactor metabolic process	0.01	Susceptible
GO:0044272	sulfur compound biosynthetic process	0.01	Susceptible
GO:0006637	acyl-CoA metabolic process	0.01	Susceptible
GO:0035383	thioester metabolic process	0.01	Susceptible
GO:0006998	nuclear envelope organization	0.01	Susceptible
GO:0006732	coenzyme metabolic process	0.01	Susceptible
GO:0042255	ribosome assembly	0.02	Susceptible
GO:0032787	monocarboxylic acid metabolic process	0.02	Susceptible
GO:0071616	acyl-CoA biosynthetic process	0.02	Susceptible
GO:0002188	translation reinitiation	0.02	Susceptible
GO:0035384	thioester biosynthetic process	0.02	Susceptible
GO:0006085	acetyl-CoA biosynthetic process	0.02	Susceptible
GO:0009107	lipoate biosynthetic process	0.02	Susceptible
GO:0009106	lipoate metabolic process	0.02	Susceptible
GO:0046364	monosaccharide biosynthetic process	0.03	Susceptible
GO:0019319	hexose biosynthetic process	0.03	Susceptible
GO:0006006	glucose metabolic process	0.03	Susceptible
GO:0006094	gluconeogenesis	0.03	Susceptible
GO:0006518	peptide metabolic process	0.03	Susceptible
GO:0042254	ribosome biogenesis	0.03	Susceptible
GO:0006412	translation	0.04	Susceptible
GO:0051188	cofactor biosynthetic process	0.04	Susceptible
GO:0006997	nucleus organization	0.04	Susceptible
GO:0015976	carbon utilization	0.05	Susceptible
GO:0010876	lipid localization	0.05	Susceptible
GO:0065009	regulation of molecular function	0.01	Resistant
GO:0050790	regulation of catalytic activity	0.02	Resistant
GO:0051716	cellular response to stimulus	0.02	Resistant
GO:0043085	positive regulation of catalytic activity	0.02	Resistant
GO:0044093	positive regulation of molecular function	0.02	Resistant
GO:0051345	positive regulation of hydrolase activity	0.04	Resistant
GO:0090304	nucleic acid metabolic process	0.04	Resistant
GO:0051336	regulation of hydrolase activity	0.05	Resistant

Note: Top section represents gene ontology biological process of predicted genes overrepresented in susceptible host background; bottom section represents biological process of predicted genes overrepresented in resistant host background. Terms arranged in ascending order by *p*-value.

**Table 5 ijms-25-02930-t005:** Information of 39 stress-related *Pst* genes among the differential expressed genes in both backgrounds.

SeqName	DE ^a^	Description	GO Names List ^b^
Resistant host background			
PSTG_14185	9.47	STE STE20 PAKA kinase	P: regulation of MAPK cascade; P: stress-activated protein kinase signaling cascade;
PSTG_11885	8.84	histone chaperone ASF1	P: positive regulation of histone acetylation; P: regulation of transcription from RNA polymerase II promoter in response to stress;
PSTG_04746	7.58	hypothetical protein	F: protein binding; P: protein import into nucleus; P: mRNA export from nucleus in response to heat stress
PSTG_03524	6.32	hypothetical protein	P: cell redox homeostasis; P: response to endoplasmic reticulum stress
PSTG_17466	6.32	hypothetical protein	P: response to stress
PSTG_07441	6.3	ABC transporter E family member 2	F: ATPase activity; P: cellular response to oxidative stress; P: translational initiation
PSTG_16290	5.37	transcription elongation factor SPT6	P: regulation of histone H3-K36 methylation; P: regulation of posttranscriptional gene silencing; P: regulation of transcription from RNA polymerase II promoter in response to stress
PSTG_01011	5.05	G2 M transition checkpoint Sum2	P: stress granule assembly; F: mRNA binding
PSTG_11190	5.05	translation initiation factor eIF-3 subunit 9	C: cytoplasmic stress granule; P: regulation of translational initiation;
PSTG_15614	5.05	hypothetical protein	F: flavin adenine dinucleotide binding; P: cellular response to oxidative stress; P: oxidation-reduction process
PSTG_06288	5.04	hypothetical protein	P: fungal-type cell wall polysaccharide biosynthetic process; P: response to salt stress; P: sphingolipid catabolic process;
PSTG_06116	4.42	hypothetical protein	F: protein disulfide isomerase activity; P: cell redox homeostasis; P: response to endoplasmic reticulum stress
PSTG_11998	4.42	AGC NDR NDR kinase	P: cellular response to osmotic stress
PSTG_06546	4.21	hypothetical protein	P: signal transduction; P: protein export from nucleus; P: cellular response to oxidative stress
PSTG_11344	3.79	hypothetical protein	P: response to oxidative stress
PSTG_12053	3.79	hypothetical protein	P: response to oxidative stress; P: cellular oxidant detoxification; P: oxidation-reduction process
PSTG_00788	3.78	hypothetical protein	F: protein tyrosine phosphatase activity; P: cellular response to oxidative stress
PSTG_04184	3.78	H3 K56 histone acetylation RTT109	P: regulation of transcription from RNA polymerase II promoter in response to stress
PSTG_09192	3.78	hypothetical protein	C: integral component of membrane; P: response to stress
PSTG_13165	3.78	peptidylprolyl isomerase	P: response to osmotic stress; P: protein peptidyl-prolyl isomerization;
PSTG_16670	3.78	AFG1-like ATPase	F: ATP binding; P: protein import into peroxisome matrix; P: cellular response to oxidative stress
PSTG_16900	3.58	STE kinase [*Puccinia graminis tritici* CRL 75-36-700-3]	F: SAM domain binding; P: invasive growth in response to glucose limitation; P: signal transduction involved in filamentous growth; P: activation of MAPKK activity; P: stress-activated protein kinase signaling cascade; P: pseudohyphal growth; P: regulation of apoptotic process
PSTG_09157	3.47	cell division cycle 14	P: cellular response to osmotic stress; C: RENT complex; F: protein tyrosine/serine/threonine phosphatase activity
PSTG_02449	3.28	translation initiation factor eIF-3 subunit 8	C: multi-eIF complex; C: cytoplasmic stress granule; F: translation initiation factor binding
PSTG_07945	3.16	hypothetical protein	F: peroxidase activity; P: response to oxidative stress; P: cellular oxidant detoxification
PSTG_09637	2.89	STE STE20 FRAY kinase	P: regulation of MAPK cascade; P: stress-activated protein kinase signaling cascade; P: regulation of apoptotic process
PSTG_07422	2.84	CMGC MAPK kinase	P: positive regulation of calcium-mediated signaling involved in cellular response to salt stress; P: peptidyl-threonine phosphorylation
PSTG_00069	2.76	STE STE20 PAKA kinase	P: regulation of MAPK cascade; P: stress-activated protein kinase signaling cascade; P: regulation of apoptotic process
PSTG_02845	2.53	glutamate decarboxylase	P: cellular response to oxidative stress; P: alanine metabolic process
PSTG_08778	2.53	translation initiation factor eIF-2	P: regulation of cytoplasmic translational initiation in response to stress
PSTG_17257	2.52	hypothetical protein	P: regulation of mRNA export from nucleus in response to heat stress; F: protein binding
PSTG_07071	2.32	ATP-dependent RNA helicase dhh1	P: stress granule assembly; C: cytoplasmic stress granule; F: protein kinase activity
PSTG_07189	2.11	cytochrome c peroxidase	P: cellular oxidant detoxification; P: cellular response to oxidative stress; P: oxidation-reduction process
PSTG_09261	2.11	regulator-nonsense transcripts 1	P: regulation of mRNA stability involved in response to oxidative stress; P: protein ubiquitination
Susceptible host			
PSTG_05154	4.75	translation initiation factor eIF-3 subunit 2	F: protein binding; P: cellular response to osmotic stress
PSTG_10795	3.96	hypothetical protein	F: protein disulfide isomerase activity; P: cell redox homeostasis; P: protein folding; P: response to endoplasmic reticulum stress
PSTG_12250	3.17	thioredoxin [*Melampsora medusae deltoidis*]	F: protein disulfide oxidoreductase activity; F: antioxidant activity; P: cell redox homeostasis; P: cellular oxidant detoxification; P: cellular response to reactive oxygen species
PSTG_20181	2.37	hypothetical protein	P: response to oxidative stress
PSTG_01910	2.37	hypothetical protein	F: ATP binding; F: unfolded protein binding; P: protein folding; P: response to stress

^a^ DE means fold of differential expression; ^b^ GO names list represents combined GO terms: cellular component (C), molecular function (F), and biological process (P).

## Data Availability

The raw datasets from the RNA-seq may be reached by request through Taras Nazarov, email: taras.nazarov@wsu.edu, or Deven R. See, email: deven_see@wsu.edu. The data are not publicly available due to the size of the files.

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
