# Peer review of "Molecular Mechanisms of the Stripe Rust Interaction with Resistant and Susceptible Wheat Genotypes"

_ijms, 2024, doi:10.3390/ijms25052930_

Round 1
Reviewer 1 Report
Comments and Suggestions for Authors
The article entitled "Molecular Mechanisms of the Stripe Rust Interaction with Resistant and Susceptible Wheat Genotypes" focuses on the interaction between wheat and the biotrophic plant pathogenic fungus Puccinia striiformis f.sp. tritici. The authors carried out transcriptome monitoring on a strain of Pst, inoculated to different wheat cultivars displaying genomic traits allowing differentiation between incompatible and compatible wheat-Pst interactions. This is very interesting work, not only in view of the data reported in the initial stages of the host-Pst interaction, but also because rusts are currently one of the most problematic diseases in wheat cultivation.
I recommend accepting this article with a few corrections, particularly in materiel and methods to fully understand what has been done.
Introduction part :
Line 8 : replace “crop production” by “wheat production”
Material en Methods part :
Paragraphe 2.1:
Please clarify, at least once, the Latin name of the wheat specie used, Triticum aestivum L. ?
Paragraphe 2.2 :
Please clarify the origin of the urediniospores used and how they were preserved. How are they multiplied? Is it also possible to get an idea of the concentration of urediniospores applied?
Is it possible to have an "average" age for plants that have reached stage 9?
Paragraphe 2.3 :
Line : 129 : using triplicated combined tissue?
If I well understand, in the line 112 you explain that you have 3 replicates to normalize for gene expression analysis…. But in fact this 3 replicates are pooled to extracted RNA ? is that what you need to understand?
You have planted 8 seeds per pot, but only 3 plants are sampled per inbred lines (21 selected) ? I didn't quite understand, 1 replicate = 1 plant or 1 pool of plants?
Figure 3 : Thank you for reviewing the quality of the figure
Reviewer 2 Report
Comments and Suggestions for Authors
The manuscript concerns molecular response of resistant and susceptible wheat varieties to stripe rust infection. The paper is well written. I have only some comments listed below:
Abstract:
Indicate % changes of gene expression, fold changes, etc. between resistant and susceptible genotypes
Introduction:
L37-44: indicate a short overview of wheat fungal pathogens and diseases. Include Fusarium spp. as one of the most problematic due to yield losses and grain contamination with mycotoxins. For this purpose refer to the following reference: https://doi.org/10.3390/agronomy13051378
Materials and Methods:
L116-117: how spores were collected? Indicate conditions for fungal growth and the type of growing medium
Results and Discussion:
Compare data for resistant and susceptible wheat genotypes in the Figures.
L323: was the annotation of genes encoding antioxidant enzymes and non-enzymatic antioxidants analyzed?
